# Improved MALDI-TOF MS Identification of *Mycobacterium tuberculosis* by Use of an Enhanced Cell Disruption Protocol

**DOI:** 10.3390/microorganisms11071692

**Published:** 2023-06-29

**Authors:** Gisele Bacanelli, Flabio Ribeiro Araujo, Newton Valerio Verbisck

**Affiliations:** 1Biotechnology and Biodiversity of the Central Western Region Postgraduate Program, Federal University of Mato Grosso do Sul, Campo Grande 79070-900, MS, Brazil; 2Embrapa Beef Cattle, Av. Rádio Maia, 830, Campo Grande 79106-550, MS, Brazil

**Keywords:** MALDI-TOF, bead beating, protein extraction, *Mycobacterium tuberculosis*

## Abstract

*Mycobacterium tuberculosis* is the microorganism that causes tuberculosis, a disease affecting millions of people worldwide. Matrix-assisted laser desorption ionization time-of-flight mass spectrometry (MALDI-TOF MS) is a fast, reliable, and cost-effective method for microorganism identification which has been used for the identification of *Mycobacterium* spp. isolates. However, the mycobacteria cell wall is rich in lipids, which makes it difficult to obtain proteins for MALDI-TOF MS analysis. In this study, two cell preparation protocols were compared: the MycoEx, recommended by MALDI-TOF instrument manufacturer Bruker Daltonics, and the MycoLyser protocol described herein, which used the MagNA Lyser instrument to enhance cell disruption with ethanol. Cell disruption and protein extraction steps with the two protocols were performed using the *Mycobacterium tuberculosis* H37Rv strain, and the MALDI-TOF MS results were compared. The MycoLyser protocol allowed for improved Biotyper identification of *M. tuberculosis* since the log(score) values obtained with this protocol were mostly ≥ 1.800 and significantly higher than that underwent MycoEx processing. The identification reliability was increased as well, considering the Bruker criteria. In view of these results, it is concluded that the MycoLyser protocol for mycobacterial cell disruption and protein extraction improves the MALDI-TOF MS method’s efficacy for *M. tuberculosis* identification.

## 1. Introduction

*Mycobacterium tuberculosis* is the microorganism that causes tuberculosis, a disease estimated to have affected 10.6 million people and caused 1.6 million deaths in 2021 [1]. Within the genus, *M. tuberculosis* is classified as a member of the *M. tuberculosis* complex (MTC), a group of closely related mycobacterial variants which also include the zoonotic pathogen *Mycobacterium bovis*, whose distinction in human and animal medical diagnosis usually requires molecular methods [2,3]. Like other mycobacteria species, *M. tuberculosis* is a rod-shaped, non-motile, non-spore-forming, and slow-growing acid-fast bacillus characterized by a high content of long-chain branched fatty acids (mycolic acids) and cord factor glycolipids in the cell wall [4].

Mass spectrometry using matrix-assisted laser desorption ionization time-of-flight (MALDI-TOF MS) for protein profile spectra characterization of microorganisms has enabled reliable identification of a large variety of bacterial species [5,6,7], including environmental and pathogenic mycobacteria [8,9,10,11], introducing reduced times and costs in clinical microbiology practice in laboratories worldwide. However, the effectiveness of MALDI-TOF MS for accurate identification of mycobacteria, including MTC, is still less satisfactory than for other bacteria. This may be explained in part by the limitations of the protein extraction procedure. The reasons for this are: (i) thickness and rigidity of the mycobacterial cell wall, which is made of a mycolyl–arabinogalactan–peptidoglycan complex structure with an outer layer of mycolic acids [4], which requires harsher methods for cell disruption and cytoplasmic protein availability; (ii) in the case of slow-growing mycobacteria, which have a lower metabolism and low number of ribosomes [12], the peak number and protein profiling are impacted; (iii) as a biosafety level 3 pathogen, mycobacteria such as *M. tuberculosis* have to undergo an inactivation step before protein extraction, which may also impact the methodological performance [10,13].

In the last few years, several protocols have proposed chemical and/or mechanical methods to address and overcome mycobacterial protein restriction for analysis. After earlier work by Saleeb [14], various other groups reported the use of silica/zirconia beads to enhance mycobacterial cell disruption and protein content release for MALDI-TOF analysis. At that time, Bruker Daltonics, a manufacturer of MALDI-TOF mass spectrometers and the Biotyper platform for microorganism identification, established the MycoEx preparation protocol, which uses beads and vortexing while acetonitrile protein extraction is performed. More recently, bead beating, which uses homogenizers with controlled timing and cycling instead of a vortex, has been shown to improve identification of mycobacteria by MALDI-TOF MS [8,10,11,15]. In previous work by our group, *M. bovis* clinical isolates could be correctly identified by MALDI-TOF MS after cell disruption with a homogenizer and protein extraction in a posterior step [15]. Nevertheless, an evaluation of those protocols, comparing, for instance, the use of a vortex and a high-performance homogenizer for mycobacterial cell disruption, has not been conducted until now.

In that context, the aim of this study was to evaluate an enhanced cell disruption protocol based on bead beating in ethanol for *Mycobacterium tuberculosis* identification by MALDI-TOF MS.

## 2. Materials and Methods

Using the virulent *Mycobacterium tuberculosis* H37Rv (NCBI NC_000962.3) as a reference strain, the MycoEx [16,17] and the MycoLyser [15] protocols for protein extraction were compared. The MycoLyser protocol used the MagNA Lyser instrument (Roche Diagnostics, Rotkreutz, Switzerland), a tissue homogenizer that simultaneously disrupts and homogenizes cells by the ultra-rapid shaking in 2 mL screw cap tubes containing beads, the applicability of which has already been demonstrated previously [18,19].

### 2.1. Mycobacterial Culture and Heat Inactivation

*M. tuberculosis* H37Rv was cultured in Löwenstein–Jensen medium by incubating at 35 °C in an ambient atmosphere for 4 weeks until enough biomass was visible on the medium. Two 10 μL inoculating loops full of bacteria were collected in 200 μL of sterile ultrapure water type I and then vortex homogenized. For inactivation, the bacterial suspension was incubated at 95 °C for 45 min, cooled to room temperature, and then 700 μL of absolute ethanol was added. The inactivation process was the same for both MycoEx and MycoLyser protocols, as described below.

### 2.2. MycoEx Protocol

After inactivation in the previous step, the samples were centrifuged for five minutes at 14,000× *g*, and the ethanol supernatant was discarded, with the pellet incubated for three minutes at room temperature to dry the remaining ethanol. For cell disruption, ~150 mg of 0.5 mm silica/zirconia beads (BioSpec Products, Bartlesville, OK, USA) were added, and homogenization was performed at maximum speed using a Vortex-Genie 2 (Scientific Industries, Inc., Bohemia, NY, USA) by vortexing for 1 min at full speed with 50 µL of acetonitrile. Then, 50 μL of formic acid was added, and proteins were retrieved in the supernatant after a 15 s vortex and two minutes of centrifugation at 14,000× *g*.

### 2.3. MycoLyser Protocol

Differently from MycoEx, cell disruption by MycoLyser was performed in ethanol. To the tube from the inactivation step, ~300 mg of 0.5 mm silica/zirconia beads (BioSpec Products) was added, and bead beating was performed in the MagNA Lyser apparatus for three cycles of 30 s at 5000 rpm. The supernatant was transferred to another tube, centrifuged for five minutes at 14,000× *g*, then discarded, and the pellet incubated for three minutes at room temperature to dry the remaining ethanol. Then, 10 μL of formic acid was added, followed by an equal volume of acetonitrile for protein extraction, which was retrieved in the supernatant after a 15 s vortex and two minutes of centrifugation at 14,000× *g*.

### 2.4. MALDI-TOF Mass Spectra Acquisition and Analysis

Proteins extracted after the MycoEx and MycoLyser protocols were applied to a MALDI-TOF target (MTP 384 ground steel, Bruker Daltonics, Billerica, MA, USA) by pipetting 1 μL of the supernatants from the protein extraction step and allowing it to air-dry at room temperature. Upon addition of 1 μL of α-cyano-4-hydroxy-cinnamic acid (5 mg/mL) in a solution containing 50% acetonitrile and 2.5% trifluoroacetic acid (*v*/*v*), crystallization of the matrix–analyte mixture was accomplished after air-drying at room temperature. A total of 18 replicates of this crystallized mixture were analyzed in an Autoflex III Smartbeam mass spectrometer (Bruker Daltonics), as described in [20]. Briefly, mass spectra in the mass range between 2000 and 20,000 Daltons were acquired with source voltage IS1 20 kV, source voltage IS2 18.55 kV, lens voltage 8.80 kV, and an ion extraction delay time of 240 ns in positive linear mode. The mass spectra of different positions in the sample-containing target were randomly obtained and summed until a value of 1.0 × 10^6^ arbitrary units was reached. The system was calibrated with a mixture of *Escherichia coli* proteins (Bacterial Test Standard, BTS, Bruker Daltonics), as recommended by the manufacturer. The list of 70 peak signals with frequencies greater than 50% was used for the generation of the main spectra profiles (MSP) using the MALDI Biotyper 3.1 program (Bruker Daltonics) with the standard configurations. For *M. tuberculosis* identification with Biotyper, the MycoEx and MycoLyser mass spectra and respective MSP were compared to the databases BDAL DB-7311 v.7.0 containing 7311 bacterial references and Mycobacterium v.5.0 with 912 mycobacterial references (Bruker Daltonics) using standard methods. The Mann–Whitney statistical test embedded in GraphPad Prism v.8 (GraphPad Software, Boston, MA, USA) was used to compare the log(score) values obtained for the protocols.

## 3. Results

When comparing the MycoEx and MycoLyser protocols for cell disruption and protein extraction using the H37Rv reference strain of *M. tuberculosis* as standard, it was observed that the highest identification log(score) with the Biotyper software was obtained after the MycoLyser protocol (Figure 1). The log(score) medians of the 18 spectra obtained experimentally were 1.656 for MycoEx and 1.862 for MycoLyser, being significantly higher after the MycoLyser protocol (*p* < 0.0001). In addition, the MycoLyser log(score) values were mostly higher than 1.800, which has been proposed as the threshold value for high-confidence mycobacterial identification at species level [10,21].

In addition to the increase in log(score) values, the identification of *M. tuberculosis* was also more accurate, as shown in Figure 2. The MSP generated by the Biotyper with the mean peak frequency of the 18 mass spectra obtained with both MycoEx and MycoLyser protocols were compared with the reference databases (BDAL plus Myco v.5.0), which contain a total of 8223 different bacterial references, 912 of which are reference spectra for mycobacteria, MTC species included. By definition, MSP is the main spectrum profile or reference spectrum of a sample, generated by the Biotyper in a previous step. It is a kind of consensus for the most reproducible spectrum found for a sample, which encompasses some tolerable variation of peaks (mass and intensity) and reflects the typical protein pattern of the sample, to be used for microorganism identification analysis. Comparing the MSP obtained for each of the protocols, MycoEx and MycoLyser, against the database revealed that the MycoLyser MSP’s first hit matched to a reference for *M. tuberculosis* (M. tuberculosis 03L LDW b) (Figure 2b), as expected, while MycoEx’s most similar match was to a *M. bovis* reference (Mycobacterium bovis Bovinus An_1 PGM) (Figure 2a).

Figure 2 shows the MycoEx and MycoLyser MSP (below, peaks in blue) mirroring at the Biotyper first hit (above, in colors), with similar (green), partially similar (yellow), or not similar (red) peaks, thus depicting the protein profile in each situation, where the correct identification for *M. tuberculosis* H37Rv was achieved with the MycoLyser protocol (Figure 2b).

Furthermore, it was found that the MycoLyser protocol provided better results regarding the accuracy of the identification since the mass spectra obtained with this protocol were more frequently identified as *M. tuberculosis*. Of the 18 MycoLyser mass spectra analyzed, 14 (78%) presented *M. tuberculosis* as the first hit (highest score), whereas for MycoEx, this was observed only for 8 mass spectra (44%). 

Moreover, the first, second, and third hits with the highest scores obtained with the MycoEx and MycoLyser mass spectra after Biotyper analysis were evaluated for identification reliability, and we verified that 11 of the 18 MycoLyser spectra (61%) had all those hits matching *M. tuberculosis*, whereas for MycoEx this was not observed.

According to Bruker Daltonics, microorganism identification reliability can be evaluated as high, medium, or low according to the consistency of the first three Biotyper hits, with the highest scores classified as being for the same genus and/or species. Those three categories were also evaluated here as follows: A = species consistency, in which the reliability of species identification is high; in this case, the three best matches must present the same identification result for genus and species; B = genus consistency, when only genus identification is highly reliable, in cases where the three best matches show the same genus identification result; C = absence of consistency for species or genus, in which neither genus nor species shows reliable identification since criteria A and B, described above, are not met. MycoLyser categorization showed 39% high (A), 44.5% medium (B), and 16.5% low (C) identification reliability, while MycoEx resulted in 0% high (A), 17% medium (B), and 83% low (C).

Interestingly, MycoLyser processing also allowed greater detection of peptides and proteins with m/z values above 8000 Daltons (Figure 2b), which may help to explain the observed higher log(score) values and greater accuracy for identification of *M. tuberculosis* H37Rv using the MycoLyser protocol.

## 4. Discussion

MALDI-TOF MS has enabled accurate and safe mycobacterial identification that is also faster and can be achieved at a lower cost [22]. However, after evaluating dozens of reports since 2010, there is still no standardized protocol for mycobacterial cell disruption and protein extraction. Several approaches have been proposed for breaking the thick cell wall of mycobacteria, including the use of detergent [23], sonication [17,24], freezing [8], and silica/zirconia beads [7], the latter being the most reported until now, as observed after a survey of the literature. Recently, a commercial kit from Bruker Daltonics demonstrated the successful inactivation of non-tuberculous mycobacteria, dispensing with the use of beads for cell disruption [9]. Although clearly allowing a faster process, the cost per sample increases with such a kit, and its chemical components are unknown.

Our work strategy focused on the analysis of cell inactivation by heat, with cell disruption and protein extraction aided by silica/zirconia beads, comparing the protocols denominated MycoLyser and MycoEx, with the latter considered a reference in the literature. We observed a significant improvement in log(score) level after MycoLyser, with 78% of spectra having log(score) values ≥ 1.8, in contrast to MycoEx, which yielded no spectra with log(score) values ≥ 1.8 (Figure 1). It is noteworthy that log(score) results were obtained without inclusion of the Biotyper databases of the MSP for the *M. tuberculosis* H37Rv sample analyzed in this study, somehow explaining the lower log(score) levels as compared to those recently reported for MTC species [10,11]. Conversely, only MycoLyser MSP resulted in the accurate identification of *M. tuberculosis* H37Rv (Figure 2b), as accessed after analysis against the Bruker Mycobacterium v.5.0 database, with 164 different species of mycobacteria [7]. Moreover, accurate identification was more frequently observed for MycoLyser (78%) than for MycoEx (44%), and only MycoLyser presented a reliability of identification with a consistent pattern, as defined by Bruker for bacterial identification at species level. 

In addition to the employment of high-performance homogenization in MycoLyser, another important difference between the protocols tested was that, while in the MycoEx protocol cell/bead vortexing was performed with acetonitrile, MycoLyser disrupted the cells after bead beating in 70% ethanol, which may help to explain its more accurate identification findings. The use of ethanol for mycobacterial inactivation and protein extraction preparation for MALDI-TOF MS was first described by Lotz [25]. Later, bioMéuriex, manufacturer of the VITEK platform for the identification of microorganisms by MALDI-TOF MS, developed a mycobacterial preparation protocol using 70% ethanol [26,27]. Interestingly, ethanol is a polar amphiphilic solvent and can extract lipids as well as disrupt the physical structure of membranes [28], which may help the waxy cell wall disruption of mycobacteria and make cellular proteins more prone to acetonitrile extraction.

In our opinion, the optimization of the protein extraction step, together with the improvement of the reference spectra databases are key points for the MALDI-TOF MS differentiation of very similar mycobacterial species, such as those belonging to the MTC. Robinne [11] has recently demonstrated MTC identification at species level and described the differentiation of *M. tuberculosis* and other MTC species with high confidence and accuracy, using a biotyping strategy with the identification of specific biomarkers for correct classification. We have also demonstrated the usefulness of this strategy for the accurate identification of *M. bovis,* another MTC species, isolated from bovine and bubaline clinical samples [15]. Since biotyping relies on peak-based discrimination for species classification [12], it is interesting to note our finding that MycoLyser has rendered many more peaks at a medium-to-high range scale in mass spectra in comparison to MycoEx. This may be an important feature for future validation of MycoLyser as a protocol for the identification of other very similar mycobacterial species or even for an antimicrobial resistance analysis of clinical cases. 

## 5. Conclusions

In this work, we have demonstrated that an enhanced mycobacterial cell disruption and protein extraction protocol, named MycoLyser, clearly improved the MALDI-TOF MS method efficacy for *M. tuberculosis* identification as compared to MycoEx, a previously established and acknowledged method. Further analysis and validation of these observations may be helpful for the establishment of mycobacterial identification at species level in clinical microbiology laboratories.

## Figures and Tables

**Figure 1 microorganisms-11-01692-f001:**
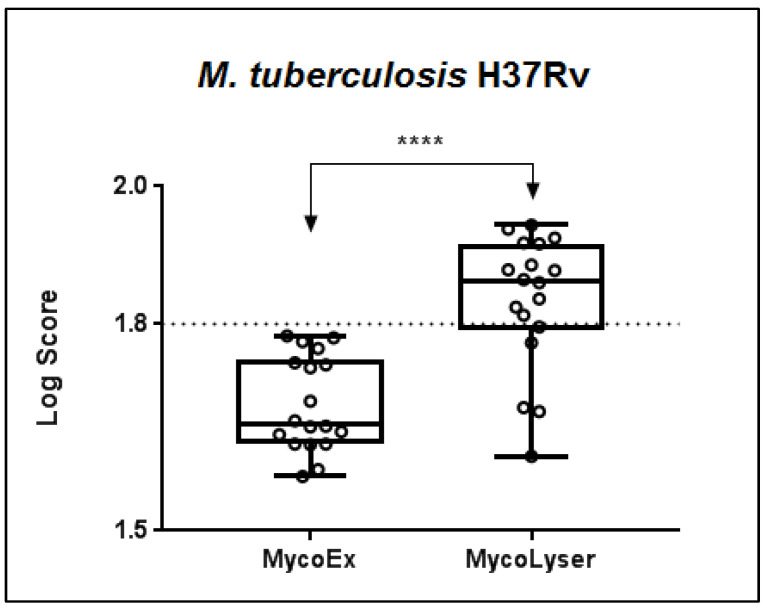
Boxplot of the MALDI Biotyper log(score) results for 18 replicate mass spectra after cell disruption and protein extraction with the MycoEx and MycoLyser protocols. Biotyper analysis was performed with BDAL DB-7311 v.7.0 plus Myco v.5.0 (Bruker Daltonics) databases with 8223 reference mass spectra; **** corresponds to a significant difference with *p* < 0.0001 (exact value, two-tailed) after Mann–Whitney test. Plotted circles represent MALDI Biotyper log(score) values and dotted line indicates log(score) 1.8 used as threshold of high confidence for mycobacteria identification at species level.

**Figure 2 microorganisms-11-01692-f002:**
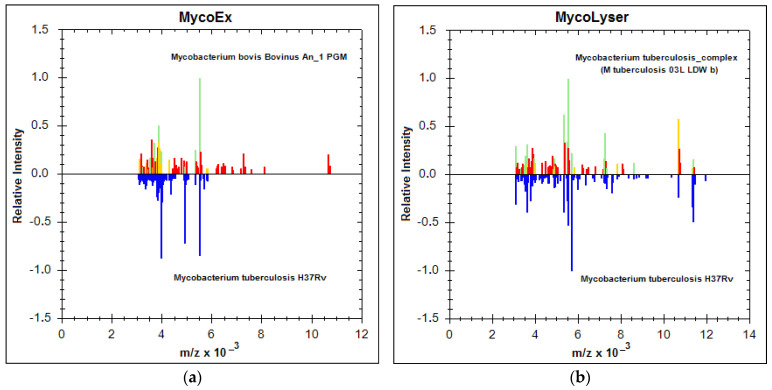
MALDI Biotyper best match identification result for the MycoEx and MycoLyser protocols. Main spectra profiles (MSP) for *Mycobacterium tuberculosis* H37Rv (below; blue peaks), generated with the 18 mass spectra after cell disruption and protein extraction with (**a**) MycoEx and (**b**) MycoLyser, are mirrored at the mass spectra (above; green, yellow and red peaks) of the respective first hit (highest score) resulting from the Biotyper analysis performed on BDAL DB-7311 v.7.0 plus Myco v.5.0 (Bruker Daltonics) databases with 8223 reference mass spectra; *m*/*z* = mass-to-charge ratio, in Daltons. Note the absence of peaks for masses above 8000 Daltons after the MycoEx extraction method.

## Data Availability

All data are available upon request.

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
