# Peer review of "Improved MALDI-TOF MS Identification of Mycobacterium tuberculosis by Use of an Enhanced Cell Disruption Protocol"

_microorganisms, 2023, doi:10.3390/microorganisms11071692_

Round 1

Reviewer 1 Report

Identification of numerous types of mycobacteria is a complex and very important methodological task. The used Matrix-Assisted Laser Desorption Ionization Time-of-Flight (MALDI-TOF MS) laser spectrometry allows to successfully identify various types of microorganisms, but it is not satisfactory enough for mycobacteria. Presented study compared two protein extraction protocols used for the identification of Mycobacterium spp. isolates. These were MycoEx, recommended by the MALDI-TOF instrument manufacturer Bruker Daltonics, and the MycoLyser protocol, which used the MagNA Lyser instrument to enhance cellular rupture in ethanol. Ethanol was used in contrast to the previously used technique. The authors focused on the analysis of cell inactivation by heat and protein extraction aided by silica/zirconia beads. As a result, they received a significant improvement in the identification of mycobacteria log (score)> 1.8. The authors have made a significant improvement in the level of identification of mycobacteria.

Author Response

Reviewer 1 comments do not point out questions to be answered.

Reviewer 2 Report

The study aimed to investigate the importance of two different protocols to enhance Mycobacterial identification using MALDI-TOF. The manuscript presents valuable information about the significance of standardizing protein isolation for mycobacterial biotyping. Although the observed trend is interesting, there are a few aspects that could be further explained. I commend the authors for their diligent efforts in exploring the differences in protein patterns isolated using the MycoEX and MycoLyser protocols.

Comments:

1.       Abstract: It is mentioned in the main manuscript that the log score should be >1.8.

2.       Line 41: The limitations discussed are primarily applicable to slow-growing mycobacteria and may not necessarily be relevant to rapid-growing mycobacterial species.

3.       Line 76: Please provide details about the type and size of beads used in the MagNA Lyser system.

4.       Figure 1: Considering that the samples were extracted using a 5-fold lower amount of solvent in the MycoLyser protocol, it would be reasonable to expect higher protein concentrations per volume and potentially improved spectra. If the authors employed any methods to normalize the protein levels for MALDI, please provide relevant details in the Methods section. Additionally, please correct the format of the numbers on the Y-axis to be consistent (e.g., 1.8 instead of 1,8).

5.       Line 135: Please define MSP in the main text for clarity.

6.       Figure 2: Please provide complete names for the axis legends. Furthermore, in Figure 2(a) title, it mentions "M. bovis Bovinus An_1 PGM," while the legend refers to Mtb H37Rv. Similarly, the title of Figure 2(b) does not match the information provided in the figure legend. It would be helpful to clarify the figure legend according to the details mentioned in lines 147-149.

7.       Line 147: Have the authors tested their protocol with other Mycobacterial species or different lineages within the Mycobacterium tuberculosis complex (MTBC)? This question is raised to gain a better understanding of why the algorithm identified M. bovis (which is closer to lineage 1 of MTB) using the spectra of H37Rv, which belongs to lineage 4.

8.       Line 162: Please clarify the relevance of gender in the context of biotyping.

9.       Line 228: It would strengthen the conclusion to include polyacrylamide gel electrophoresis (PAGE) gels to compare the types of proteins isolated using the two different methods discussed in this paper.

Author Response

Regarding the differences between the protein profiles (patterns), yes, they are different after cell disruption with the two methods (MycoEx and MycoLyser). This is clear after checking the mass lists and finding different peaks, probably corresponding to different peptides and/or proteins. Instead of showing complete mass lists we decided to depict main differences in figure 2, which shows the protein profiles. 
Although it might be interesting to further explore this, such as for investigation of antimicrobial resistance or virulence profiles, the fact is that this was not our focus at this time. For the purpose of identification of microorganisms, using the Biotyper or anyother software, the information presented is sufficient. Once the reference mass spectra/main spectra profile (MSP) have been generated, with the two methods MycoEx and MycoLyser, for the same reference sample, M. tuberculosis H37Rv, MSPs confrontation with the database can be directly evaluated, as presented in figure 2. The observed differences are suggested to support M. tuberculosis improvement of identification accuracy, as reported in the results section.

1.       Abstract: It is mentioned in the main manuscript that the log score should be >1.8.

Answer to comment 1: Right, it should be 1.8, not 1,8. It was wrongly typed and will be corrected in Abstract.

2.       Line 41: The limitations discussed are primarily applicable to slow-growing mycobacteria and may not necessarily be relevant to rapid-growing mycobacterial species.

Answer to comment 2: Okay, that's a relevant observation, but there doesn't seem to be in conflict with the wording of the section on lines 40-50, because fast-growing mycobacteria also present problems for protein analysis by MALDI-TOF, as reported in the literature, please see reference 8 (Rodriguez-Temporal et al., 2018). Anyway, the text was rearranged from generic to specific as follows:

"However, the effectiveness of MALDI-TOF MS for accurate identification of mycobacteria, including MTC, is still less satisfactory than for other bacteria. This may be explained in part by limitations of protein extraction procedure. Reasons for that are: i) thickness and rigidity of the mycobacterial cell wall, made of a mycolyl-arabinogalactan-peptidoglycan complex structure with an outer layer of mycolic acids [4], what requires harsher methods for cell disruption and cytoplasmic protein availability; ii) in case of slow-growing mycobacteria, which have a lower metabolism and low number of ribosomes [12], peak number and protein profiling are impacted; iii) as a biosafety level 3 pathogen, mycobacteria such as M. tuberculosis have to undergo an inactivation step before protein extraction, which may also impact the methodological performance [10,13]."

3.       Line 76: Please provide details about the type and size of beads used in the MagNA Lyser system.

Answer to comment 3: Beads were the same for MycoEx and MycoLyser methods, and its type and size, as well as manufacturer, are mentioned in lines 92 and 100 of main text at Materials and Methods section.

4.       Figure 1: Considering that the samples were extracted using a 5-fold lower amount of solvent in the MycoLyser protocol, it would be reasonable to expect higher protein concentrations per volume and potentially improved spectra. If the authors employed any methods to normalize the protein levels for MALDI, please provide relevant details in the Methods section. Additionally, please correct the format of the numbers on the Y-axis to be consistent (e.g., 1.8 instead of 1,8).

Answer to comment 4: MycoEx was performed as reported by Bruker. We started with the same amount of cells for both protocols. Protein normalization was not performed in our evaluation because MALDI-TOF is not a quantitative analytical technique. However, we established 1x10e6 arbitrary units as the criterion for a mass spectra acquisition for both MycoEx and MycoLyser and this should be enough to remove any bias regarding MALDI sensitivity. So, it is quite reasonable the observed differences in protein profiles are due to the processes of cell disruption.

Numbering in Figure 1 Y-axis will also be fixed.

5.       Line 135: Please define MSP in the main text for clarity.

Answer to comment 5: MSP definition will be included in paragraph "MALDI-TOF analysis", at Materials and Methods section, and mentioned again in line 140 of Results section.

6.       Figure 2: Please provide complete names for the axis legends. Furthermore, in Figure 2(a) title, it mentions "M. bovis Bovinus An_1 PGM," while the legend refers to Mtb H37Rv. Similarly, the title of Figure 2(b) does not match the information provided in the figure legend. It would be helpful to clarify the figure legend according to the details mentioned in lines 147-149.

Answer to comment 6: Figure 2 titles and legends will be reviewed to make it clear. Title in Figure 2(a) and 2(b) will be changed to "MycoEx" and "MycoLyser", respectively. X-axis and Y-axis legend will be typed "Relative Intensity" and "m/z (10e-3)", respectively, and m/z will be explained in legend as to be mass-to-charge ratio.

7. Line 147: Have the authors tested their protocol with other Mycobacterial species or different lineages within the Mycobacterium tuberculosis complex (MTBC)? This question is raised to gain a better understanding of why the algorithm identified M. bovis (which is closer to lineage 1 of MTB) using the spectra of H37Rv, which belongs to lineage 4.

Answer to comment 7: We have also analyzed M. bovis isolates, with both protocols MycoEx and MycoLyser, as reported earlier. We have not tested other MTBC species or lineages. Database used for identification was from Bruker, including reference spectra for 912 samples of mycobacteria processed with beads. We have not included to that database, prior to analysis, our own MSPs, obtained with MycoEx and MycoLyser, so, apart from any differences between our M. tuberculosis H37Rv sample and other M. tuberculosis isolates used by Bruker, regarding geographic origin, culture conditions, etc, we would expect that identification result would be reflecting the differences in protein profiles obtained with the two cell disruption protocols. Bruker never proposed that its MycoEx protocol and Mycobacterial libraries would identify MTBC species or lineages, but maybe this could be achievable with our MycoLyser protocol. 

8.       Line 162: Please clarify the relevance of gender in the context of biotyping.

Answer to comment 8: Right, it should be genus, not gender. It was wrongly typed and will be corrected.

9.       Line 228: It would strengthen the conclusion to include polyacrylamide gel electrophoresis (PAGE) gels to compare the types of proteins isolated using the two different methods discussed in this paper.

Answer to comment 9: We will review our conclusion to "In this work, we have demonstrated that an enhanced cell disruption protocol, named MycoLyser, improved M. tuberculosis identification by MALDI-TOF MS, in comparison to a previously established and acknowledged method. Further analysis and validation of these observations may be helpful for the establishment of mycobacterial identification at species level in clinical microbiology laboratories."